# Alternators For Sequence Modeling

**Mohammad R. Rezaei**                                                   *mr.rezaei@mail.utoronto.ca*
*University of Toronto*
*Vertaix*

**Adji Bousso Dieng**                                                   *adji@princeton.edu*
*Department of Computer Science, Princeton University*
*Vertaix*

**Reviewed on OpenReview:** *https://openreview.net/forum?id=Q7OC1HQOVO*

## Abstract

This paper introduces *alternators*, a novel family of non-Markovian dynamical models for sequences. An alternator features two neural networks: the *observation trajectory network* (OTN) and the *feature trajectory network* (FTN). The OTN and the FTN work in conjunction, alternating between outputting samples in the observation space and some feature space, respectively. The parameters of the OTN and the FTN are not time-dependent and are learned via a minimum cross-entropy criterion over the trajectories. Alternators are versatile. They can be used as dynamical latent-variable generative models or as sequence-to-sequence predictors. Alternators can uncover the latent dynamics underlying complex sequential data, accurately forecast and impute missing data, and sample new trajectories. We showcase the capabilities of alternators in three applications. We first used alternators to model the Lorenz equations, often used to describe chaotic behavior. We then applied alternators to Neuroscience, to map brain activity to physical activity. Finally, we applied alternators to Climate Science, focusing on sea-surface temperature forecasting. In all our experiments, we found alternators are stable to train, fast to sample from, yield high-quality generated samples and latent variables, and often outperform strong baselines such as Mambas, neural ODEs, and diffusion models in the domains we studied.

## 1 Introduction

Time underpins many scientific processes and phenomena. These are often modeled using differential equations (Schrödinger, 1926; Lorenz, 1963; McLean, 2012). Developing these equations requires significant domain knowledge. Over the years, scientists have developed various families of differential equations for modeling specific classes of problems. The interpretability of these equations makes them appealing. However, differential equations are often intractable. Numerical solvers have been developed to find approximate solutions, often with significant computation overhead (Wanner & Hairer, 1996; Hopkins & Furber, 2015). Several works have leveraged neural networks to speed up or replace numerical solvers. For example, neural operators have been developed to approximately solve differential equations (Kovachki et al., 2023). Neural operators extend traditional neural networks to operate on functions instead of fixed-size vectors. They can approximate solutions to complex functional relationships described as partial differential equations. However, neural operators still require data from numerical solvers to train their neural networks. They may face challenges in generalizing to unseen data and are sensitive to hyperparameters (Li et al., 2021; Kontolati et al., 2023).

Beyond their intractability, differential equations as a framework may not be amenable to all time-dependent problems. For example, it is not clear how to model language, which is inherently sequential, using differential equations. For such general problems that are inherently time-dependent, fully data-driven methods become appealing. These methods are faced with the complexities that time-dependent data often exhibit, including

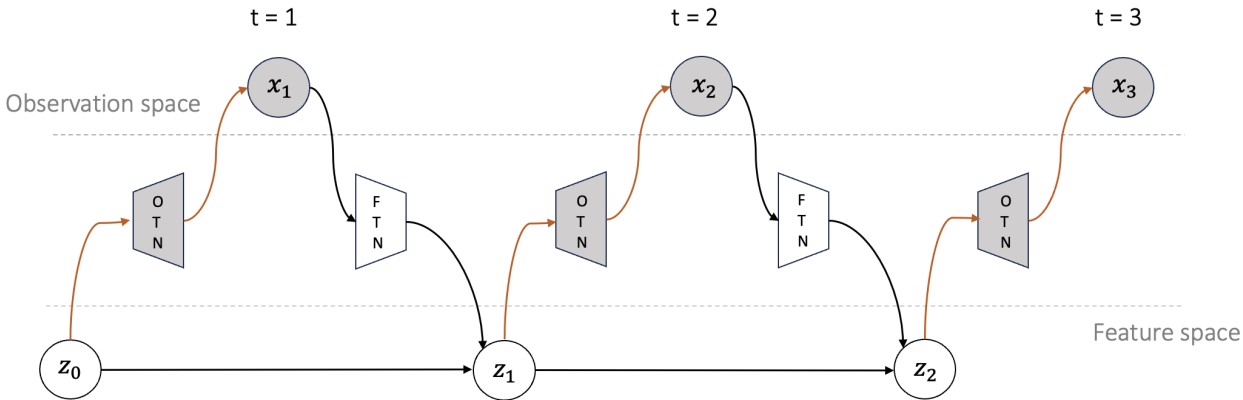

Figure 1: Generative process of an alternator with a cycle of length $T = 3$. An initial random feature $\boldsymbol{z}_0$ is generated from a fixed distribution, e.g. a standard Gaussian. The rest of the observations $\boldsymbol{x}_{1:T}$ and features $\boldsymbol{z}_{1:T}$ are generated by alternating between sampling from the OTN and the FTN, respectively.

high stochasticity, high dimensionality, and nontrivial temporal dependencies. Generative modeling is a data-driven framework that has been widely used to model sequences. Several dynamical generative models have been proposed over the years (Gregor et al., 2014; Fraccaro et al., 2016; Du et al., 2016; Dieng et al., 2016; 2019; Kobyzev et al., 2020; Ho et al., 2020; Kobyzev et al., 2020; Rasul et al., 2021; Yan et al., 2021; Dutordoir et al., 2022; Li et al., 2022b; Neklyudov et al., 2022; Lin et al., 2023; Li et al., 2024). Unlike differential equations, generative models can account for the stochasticity in observations and can be easy to generate data from. However, they are less interpretable than differential equations, may require significant training data, and often fail to produce predictions and samples that are faithful to the underlying dynamics.

This paper introduces *alternators*, a new framework for modeling time-dependent data. Alternators model dynamics using two neural networks called the observation trajectory network (OTN) and the feature trajectory network (FTN), that alternate between generating observations and features over time, respectively. These two neural networks are fit by minimizing the cross entropy of two joint distributions defined over the observation and feature trajectories. This framework offers great flexibility. Alternators can be used as generative models, in which case the features correspond to interpretable low-dimensional latent variables that capture the hidden dynamics governing the observed sequences. Alternators can also be used to map an observed sequence to an associated observed sequence, for supervised learning. In this case, the features represent low-dimensional representations of the input sequences. These features are then used to predict the output sequences. Alternators can be used to efficiently impute missing data, forecast, sample new trajectories, and encode sequences. Figure 1 illustrates the generative process of an alternator over three time steps.

Section 4 showcases the capabilities of alternators in three different applications: the Lorenz attractor, neural decoding of brain activity, and sea-surface temperature forecasting. In all these applications, we found that alternators tend to outperform other sequence models, including dynamical VAEs, neural ODEs, diffusion models, and Mambas, on different sequence modeling tasks.

Alternators present significant opportunities and considerations across multiple domains. In scientific applications, the framework's ability to model complex temporal dynamics with interpretable low-dimensional latent variables could accelerate discovery in fields ranging from climate science to neuroscience, where understanding underlying dynamical systems is crucial. Our contributions are threefold: (1) introducing Alternators, a novel framework using alternating OTN/FTN networks trained via cross-entropy minimization, (2) demonstrating versatility across generative modeling, prediction, imputation, and forecasting with minimal modifications, and (3) showing consistent superior performance across dynamical systems.

## 2   Alternators

We are interested in modeling time-dependent data in a general and flexible way. We seek to be able to sample new plausible sequences fast, impute missing data, forecast the future, learn the dynamics underlying observed sequences, learn good low-dimensional representations of observed sequences, and accurately predict sequences. We now describe *alternators*, a framework for modeling sequences that offers the capabilities described above.

**Generative Modeling.**   We assume the data are from an unknown sequence distribution, which we denote by $p(\boldsymbol{x}_{1:T})$, with $T$ being a pre-specified sequence length. Here each $\boldsymbol{x}_t \in \mathbb{R}^{D_x}$. We approximate $p(\boldsymbol{x}_{1:T})$ with a model with the following generative process:

1. Sample $\boldsymbol{z}_0 \sim p(\boldsymbol{z}_0)$.

2. For $t = 1, \ldots, T$:

   (a) Sample $\boldsymbol{x}_t \sim p_\theta(\boldsymbol{x}_t \,|\, \boldsymbol{z}_{t-1})$.
   (b) Sample $\boldsymbol{z}_t \sim p_\phi(\boldsymbol{z}_t \,|\, \boldsymbol{z}_{t-1}, \boldsymbol{x}_t)$ .

Here $\boldsymbol{z}_{0:T}$ is a sequence of low-dimensional latent variables that govern the observation dynamics. Each $\boldsymbol{z}_t \in \mathbb{R}^{D_z}$, with $D_z \ll D_x$. The distribution $p(\boldsymbol{z}_0)$ is a prior over the initial latent variable $\boldsymbol{z}_0$. It is fixed. The distributions $p_\theta(\boldsymbol{x}_t \,|\, \boldsymbol{z}_{t-1})$ and $p_\phi(\boldsymbol{z}_t \,|\, \boldsymbol{z}_{t-1}, \boldsymbol{x}_t)$ relate the observations and the latent variables at each time step. They are parameterized by $\theta$ and $\phi$, which are unknown. The latent variable $\boldsymbol{z}_{t-1}$ acts as a dynamic memory used to predict the next observation $\boldsymbol{x}_t$ at time $t$ and to update its state to $\boldsymbol{z}_t$ using the newly observed $\boldsymbol{x}_t$.

The generative process described above induces a valid joint distribution over the data trajectory $\boldsymbol{x}_{1:T}$ and the latent trajectory $\boldsymbol{z}_{0:T}$,

$$p_{\theta,\phi}(\boldsymbol{x}_{1:T}, \boldsymbol{z}_{0:T}) = p(\boldsymbol{z}_0) \prod_{t=1}^{T} p_\theta(\boldsymbol{x}_t \,|\, \boldsymbol{z}_{t-1}) p_\phi(\boldsymbol{z}_t \,|\, \boldsymbol{z}_{t-1}, \boldsymbol{x}_t). \tag{1}$$

This joint yields valid marginals over the latent trajectory and data trajectory,

$$p_{\theta,\phi}(\boldsymbol{x}_{1:T}) = \int \left\{ p(\boldsymbol{z}_0) \prod_{t=1}^{T} p_\theta(\boldsymbol{x}_t \,|\, \boldsymbol{z}_{t-1}) p_\phi(\boldsymbol{z}_t \,|\, \boldsymbol{z}_{t-1}, \boldsymbol{x}_t) \right\} d\boldsymbol{z}_{0:T} \tag{2}$$

$$p_{\theta,\phi}(\boldsymbol{z}_{0:T}) = \int \left\{ p(\boldsymbol{z}_0) \prod_{t=1}^{T} p_\theta(\boldsymbol{x}_t \,|\, \boldsymbol{z}_{t-1}) p_\phi(\boldsymbol{z}_t \,|\, \boldsymbol{z}_{t-1}, \boldsymbol{x}_t) \right\} d\boldsymbol{x}_{1:T} \tag{3}$$

These two marginals describe flexible models over the data and latent trajectories. Even though the model is amenable to any distribution, here we describe distributions for modeling continuous data. We define

$$p(\boldsymbol{z}_0) = \mathcal{N}(0, \boldsymbol{I}) \tag{4}$$

$$p_\theta(\boldsymbol{x}_t \,|\, \boldsymbol{z}_{t-1}) = \mathcal{N}\left(\sqrt{(1-\sigma_x^2)} \cdot f_\theta(\boldsymbol{z}_{t-1}), \ D_x \sigma_x^2\right) \tag{5}$$

$$p_\phi(\boldsymbol{z}_t \,|\, \boldsymbol{z}_{t-1}, \boldsymbol{x}_t) = \mathcal{N}\left(\sqrt{\alpha_t} \cdot g_\phi(\boldsymbol{x}_t) + \sqrt{(1-\alpha_t-\sigma_z^2)} \cdot \boldsymbol{z}_{t-1}, \ D_z \sigma_z^2\right), \tag{6}$$

where $f_\theta(\cdot)$ and $g_\phi(\cdot)$ are two neural networks, called the observation trajectory network (OTN) and the feature trajectory network (FTN), respectively. Here $\sigma_x^2$ and $\sigma_z^2$ are hyperparameters such that $\sigma_z^2 < \sigma_x^2$. The sequence $\alpha_{1:T}$ is also fixed and pre-specified. Each $\alpha_t$ is such that $0 \leq \alpha_t \leq 1 - \sigma_z^2$.

**Learning.**   Traditionally, latent-variable models such as the one described above are learned using variational inference (Blei et al., 2017). Here we proceed differently and fit alternators by minimizing the cross-entropy between the joint distribution defining the model $p_{\theta,\phi}(\boldsymbol{x}_{1:T}, \boldsymbol{z}_{0:T})$ and the joint distribution defined as the

product of the marginal distribution over the latent trajectories $p_{\theta,\phi}(\boldsymbol{z}_{0:T})$ and the data distribution $p(\boldsymbol{x}_{1:T})$. That is, we learn the model parameters $\theta$ and $\phi$ by minimizing the following objective:

$$\mathcal{L}(\theta,\phi) = -\mathbb{E}_{p(\boldsymbol{x}_{1:T})\cdot p_{\theta,\phi}(\boldsymbol{z}_{0:T})}\left[\log p_{\theta,\phi}(\boldsymbol{x}_{1:T}, \boldsymbol{z}_{0:T})\right]. \tag{7}$$

To gain more intuition on why minimizing $\mathcal{L}(\theta,\phi)$ is a good thing to do, let's expand it using Bayes' rule,

$$\mathcal{L}(\theta,\phi) = -\mathbb{E}_{p(\boldsymbol{x}_{1:T})\cdot p_{\theta,\phi}(\boldsymbol{z}_{0:T})}\left[\log p_{\theta,\phi}(\boldsymbol{z}_{0:T}) + \log p_{\theta,\phi}(\boldsymbol{x}_{1:T} \mid \boldsymbol{z}_{0:T})\right] \tag{8}$$

$$= \mathcal{H}(p_{\theta,\phi}(\boldsymbol{z}_{0:T})) + \mathbb{E}_{p_{\theta,\phi}(\boldsymbol{z}_{0:T})}\left[\mathrm{KL}(p(\boldsymbol{x}_{1:T})\|p_{\theta,\phi}(\boldsymbol{x}_{1:T} \mid \boldsymbol{z}_{0:T}))\right]. \tag{9}$$

Here $\mathcal{H}(p_{\theta,\phi}(\boldsymbol{z}_{0:T}))$ is the entropy of the marginal over the latent trajectory and the second term is the expected Kullback-Leibler (KL) divergence between the data distribution $p(\boldsymbol{x}_{1:T})$ and $p_{\theta,\phi}(\boldsymbol{x}_{1:T} \mid \boldsymbol{z}_{0:T})$, the conditional distribution of the data trajectory given the latent trajectory.

Eq. 9 is illuminating. Indeed, it says that minimizing $\mathcal{L}(\theta,\phi)$ with respect to $\theta$ and $\phi$ minimizes the entropy of the marginal over the latent trajectory, which maximizes the information gain on the latent trajectories. This leads to *good* latent representations. On the other hand, minimizing $\mathcal{L}(\theta,\phi)$ also minimizes the expected KL between the data distribution and the conditional distribution of the observed sequence given the latent trajectory. This forces the OTN to learn parameter settings that generate *plausible* sequences and forces the FTN to generate latent trajectories that yield good data trajectories.

It may be tempting to view Eq. 9 as the *evidence lower bound* (ELBO) objective function optimized by a variational autoencoder (VAE) (Kingma et al., 2019). That would be incorrect for two reasons. First, interpreting Eq. 9 as an ELBO would require interpreting $p_{\theta,\phi}(\boldsymbol{z}_{0:T})$ as an approximate posterior distribution, or a variational distribution, which we can't do since $p_{\theta,\phi}(\boldsymbol{z}_{0:T})$ depends explicitly on model parameters. Second, Eq. 9 is the sum of the entropy of $p_{\theta,\phi}(\boldsymbol{z}_{0:T})$ and the expected log-likelihood of the observed sequence, whereas an ELBO would have been the sum of the entropy of the variational distribution and the expected log-joint of the observed sequence and the latent trajectory.

To minimize $\mathcal{L}(\theta,\phi)$ we expand it further using the specific distributions we defined in Eq. 4, 5, and 6,

$$\mathcal{L}(\theta,\phi) = \mathbb{E}_{p(\boldsymbol{x}_{1:T})\cdot p_{\theta,\phi}(\boldsymbol{z}_{0:T})}\left[\sum_{t=1}^{T}\|\boldsymbol{z}_t - \boldsymbol{\mu}_{z_t}\|_2^2 + \frac{D_z\sigma_z^2}{D_x\sigma_x^2}\cdot\|\boldsymbol{x}_t - \boldsymbol{\mu}_{x_t}\|_2^2\right] \tag{10}$$

$$\boldsymbol{\mu}_{x_t} = \sqrt{(1-\sigma_x^2)}\cdot f_\theta(\boldsymbol{z}_{t-1}) \text{ and } \boldsymbol{\mu}_{z_t} = \sqrt{\alpha_t}\cdot g_\phi(\boldsymbol{x}_t) + \sqrt{(1-\alpha_t-\sigma_z^2)}\cdot \boldsymbol{z}_{t-1} \tag{11}$$

Although Eq. 10 is intractable–it still depends on expectations–we can approximate it using Monte Carlo,

$$\mathcal{L}(\theta,\phi) \approx \frac{1}{B}\sum_{b=1}^{B}\sum_{t=1}^{T}\left[\left\|\boldsymbol{z}_t^{(b)} - \boldsymbol{\mu}_{z_t^{(b)}}\right\|_2^2 + \frac{D_z\sigma_z^2}{D_x\sigma_x^2}\cdot\left\|\boldsymbol{x}_t^{(b)} - \boldsymbol{\mu}_{x_t^{(b)}}\right\|_2^2\right], \tag{12}$$

where $\boldsymbol{x}_{1:T}^{(1)},\ldots,\boldsymbol{x}_{1:T}^{(B)}$ are data trajectories sampled from the data distribution[1] and $\boldsymbol{z}_{0:T}^{(1)},\ldots,\boldsymbol{z}_{0:T}^{(B)}$ are latent trajectories sampled from the marginal $p_{\theta,\phi}(\boldsymbol{z}_{0:T})$ using ancestral sampling on Eq. 3.

Algorithm 1 summarizes the procedure for dynamical generative modeling with alternators. At each time step $t$, the OTN tries to produce its best guess for the observation $\boldsymbol{x}_t$ using the current memory $\boldsymbol{z}_{t-1}$. The output from the OTN is then passed as input to the FTN to update the dynamic memory from $\boldsymbol{z}_{t-1}$ to $\boldsymbol{z}_t$. This update is modulated by $\alpha_t$, which determines how much we rely on the memory $\boldsymbol{z}_{t-1}$ compared to the new observation $\boldsymbol{x}_t$. When dealing with data sequences for which we know certain time steps correspond to more noisy observations than others, we can use $\alpha_t$ to rely more on the memory $\boldsymbol{z}_{t-1}$ than the noisy observation $\boldsymbol{x}_t$. When the noise in the observed sequences is not known, which is often the case, we set $\alpha_t$ fixed across time. The ability to change $\alpha_t$ across time steps provides alternators with an enhanced ability to handle noisy observations compared to other generative modeling approaches to sequence modeling.

---

[1]Although the true data distribution $p(\boldsymbol{x}_{1:T})$ is unknown, we have some samples from it which are the observed sequences, which we can use to approximate the expectation.

---

**Algorithm 1:** Dynamical Generative Modeling with Alternators

---

Inputs: Samples from $p(\boldsymbol{x}_{1:T})$, batch size $B$, variances $\sigma_x^2$ and $\sigma_z^2$, schedule $\alpha_{1:T}$
Initialize model parameters $\theta$ and $\phi$
**while** *not converged* **do**
    **for** $b = 1, \ldots, B$ **do**
        Draw initial latent $\boldsymbol{z}_0^{(b)} \sim \mathcal{N}(0, I_{D_z})$
        **for** $t = 1, \ldots, T$ **do**
            Draw noise variables $\boldsymbol{\epsilon}_{xt}^{(b)} \sim \mathcal{N}(0, I_{D_x})$ and $\boldsymbol{\epsilon}_{zt}^{(b)} \sim \mathcal{N}(0, I_{D_z})$
            Draw $\boldsymbol{x}_t^{(b)} = \sqrt{(1-\sigma_x^2)} \cdot f_\theta(\boldsymbol{z}_{t-1}^{(b)}) + \sigma_x \cdot \boldsymbol{\epsilon}_{xt}^{(b)}$
            Draw $\boldsymbol{z}_t^{(b)} = \sqrt{\alpha_t} \cdot g_\phi(\boldsymbol{x}_t^{(b)}) + \sqrt{(1-\alpha_t-\sigma_z^2)} \cdot \boldsymbol{z}_{t-1}^{(b)} + \sigma_z \cdot \boldsymbol{\epsilon}_{zt}^{(b)}$
        **end**
    **end**
    Compute loss $\mathcal{L}(\theta, \phi)$ in Eq. 12 using $\boldsymbol{z}_{0:T}^{1:B}$ and data samples from $p(\boldsymbol{x}_{1:T})$
    Backpropagate to get $\nabla_\theta \mathcal{L}(\theta, \phi)$ and $\nabla_\phi \mathcal{L}(\theta, \phi)$
    Update parameters $\theta$ and $\phi$ using stochastic optimization, e.g. Adam.
**end**

---

**Sequence-To-Sequence Prediction.** When given paired sequences $\boldsymbol{x}_{1:T}$ and $\boldsymbol{y}_{1:T}$, we can use alternators to predict $\boldsymbol{y}_{1:T}$ given $\boldsymbol{x}_{1:T}$ and vice-versa. We simply replace $p(\boldsymbol{x}_{1:T})p_{\theta,\phi}(\boldsymbol{z}_{0:T})$ with the product of the joint data distribution, $p(\boldsymbol{x}_{1:T}, \boldsymbol{y}_{1:T})$ and $p(\boldsymbol{z}_0)$. The objective remains the cross entropy,

$$\mathcal{L}(\theta, \phi) = -\mathbb{E}_{p(\boldsymbol{x}_{1:T}, \boldsymbol{y}_{1:T})p(\boldsymbol{z}_0)}\left[\log p_{\theta,\phi}(\boldsymbol{x}_{1:T}, \boldsymbol{y}_{1:T}, \boldsymbol{z}_0)\right]. \tag{13}$$

This leads to the same tractable objective as Eq. 12, replacing $\boldsymbol{z}_{1:T}$ with $\boldsymbol{y}_{1:T}$,

$$\mathcal{L}(\theta, \phi) \approx \frac{1}{B}\sum_{b=1}^{B}\sum_{t=1}^{T}\left[\left\|\boldsymbol{y}_t^{(b)} - \boldsymbol{\mu}_{y_t^{(b)}}\right\|_2^2 + \frac{D_y \sigma_y^2}{D_x \sigma_x^2} \cdot \left\|\boldsymbol{x}_t^{(b)} - \boldsymbol{\mu}_{x_t^{(b)}}\right\|_2^2\right], \tag{14}$$

where $\boldsymbol{x}_{1:T}^{(1)}, \ldots, \boldsymbol{x}_{1:T}^{(B)}$ and $\boldsymbol{y}_{1:T}^{(1)}, \ldots, \boldsymbol{y}_{1:T}^{(B)}$ are sequence pairs sampled from the data distribution, $\boldsymbol{\mu}_{x_t} = \sqrt{(1-\sigma_x^2)} \cdot f_\theta(\boldsymbol{y}_{t-1})$ and $\boldsymbol{\mu}_{y_t} = \sqrt{\alpha_t} \cdot g_\phi(\boldsymbol{x}_t) + \sqrt{(1-\alpha_t-\sigma_y^2)} \cdot \boldsymbol{y}_{t-1}$. Algorithm 2 summarizes the procedure for sequence-to-sequence prediction with alternators.

**Imputation and forecasting.** Imputing missing values and forecasting future events are simple using alternators. We simply follow the generative process of an alternator, each time using $\boldsymbol{x}_t$ when it is observed or sampling it from $p_\theta(\boldsymbol{x}_t \mid \boldsymbol{z}_{t-1})$ when it is missing.

**Encoding sequences.** It is easy to get a low-dimensional sequential representation of a new sequence $\boldsymbol{x}_{1:T}^*$: we simply plug $x_t^*$ at each time step $t$ in the mean of the distribution $p_\phi(\boldsymbol{z}_t \mid \boldsymbol{z}_{t-1}, \boldsymbol{x}_t)$ in Eq. 6,

$$\boldsymbol{z}_t^* = \sqrt{\alpha_t} \cdot g_\phi(\boldsymbol{x}_t^*) + \sqrt{(1-\alpha_t-\sigma_z^2)} \cdot \boldsymbol{z}_{t-1}^*. \tag{15}$$

The sequence $\boldsymbol{z}_{1:T}^*$ is the low-dimensional representation of $\boldsymbol{x}_{1:T}^*$ given by the alternator. To uncover the dynamics underlying a collection of $B$ sequences $\boldsymbol{x}_{1:T}^{(1)*}, \ldots, \boldsymbol{x}_{1:T}^{(B)*}$ instead, we can simply use Eq. 15 for each sequence $\boldsymbol{x}_{1:T}^{(b)*}$ and take the mean for each time step. The resulting sequence is a compact representation of the dynamics governing the input sequences.

## 3 Related Work

Alternators are a new family of models for time-dependent data. As such, they are related to many existing dynamical models.

Autoregressive models (ARs) define a probability distribution for the next element in a sequence based on the previous elements, making them effective for modeling high-dimensional structured data (Gregor et al.,

2014). They have been widely used in applications such as speech recognition Chung et al. (2019), language modeling Black et al. (2022), and image generative modeling Chen et al. (2018b). However, ARs don't have latent variables, which may limit their applicability.

Temporal point processes (TPPs) were introduced to model event data (Du et al., 2016). TPPs model both event timings and associated markers by defining an intensity function that is a nonlinear function of history using recurrent neural networks (RNNs). However, TPPs lack latent variables and are only amenable to discrete data, which limits their applicability.

Dynamical VAEs such as VRNNs (Chung et al., 2015) and SRNNs (Fraccaro et al., 2016) model sequences by parameterizing VAEs with RNNs and bidirectional RNNs, respectively. This enables these methods to learn good representations of time-dependent data by maximizing the ELBO. However, they fail to generalize and struggle with generating good observations due to their parameterizations of the sampling process.

Differential equations are the traditional way dynamics are modeled in the sciences. However, they may be slow to resolve. Recently, neural operators have been developed to extend traditional neural networks to operate on functions instead of fixed-size vectors (Kovachki et al., 2023). They can approximate solutions to complex functional relationships modeled as partial differential equations. However, neural operators rely on numerical solvers to train their neural networks. They may struggle to generalize to unseen data and are sensitive to hyperparameters (Li et al., 2021; Kontolati et al., 2023).

Neural ordinary differential equations (NODEs) model time-dependent data using a neural network to predict an initial latent state which is then used to initialize a numerical solver that produces trajectories (Chen et al., 2018a). NODEs enables continuous-time modeling of complex temporal patterns. They provide a more flexible framework than traditional ODE solvers for modeling time series data. However, NODEs are still computationally costly and can be challenging to train since they require careful tuning of hyperparameters and still rely on numerical solvers to ensure stability and convergence (Finlay et al., 2020). Furthermore, NODEs are deterministic; stochasticity in NODEs is only modeled in the initial state. This makes NODEs not ideal for modeling noisy observations. Alternators also differ fundamentally from neural stochastic differential equations (SDEs) introduced by Liu et al. (2019) in several key ways. While neural SDEs incorporate stochastic terms to model randomness, but rely on computationally expensive numerical solvers and maintain high-dimensional state representations, Alternators use direct neural network mappings and explicitly model low-dimensional latent variables ($D_z \ll D_x$), making them significantly faster as demonstrated in our sea-surface temperature forecasting experiments while maintaining stochasticity through noise models for both latent and observation spaces.

Probability flows are generative models that utilize invertible transformations to convert simple base distributions into complex, multimodal distributions (Kobyzev et al., 2020). They employ continuous-time stochastic processes to model dynamics. These models explicitly represent probability distributions using normalizing flow (Papamakarios et al., 2021). While normalizing flows offer advantages such as tractable computation of log-likelihoods, they have high-dimensional latent variables and require invertibility, which hinders flexibility.

Recently, diffusion models have been used to model sequences (Lin et al., 2023). For example DDPMs can be used to denoise a sequence of noise-perturbed data by iteratively removing the noise from the sequence (Rasul et al., 2021; Yan et al., 2021; Biloš et al., 2022; Lim et al., 2023). This iterative refinement enables DDPMs to generate high-quality samples. TimeGrad is a diffusion-based approach that introduces noise at each time step and gradually denoises it through a backward transition kernel conditioned on historical time series (Rasul et al., 2021). ScoreGrad follows a similar strategy but extends the diffusion process to a continuous domain, replacing discrete steps with interval-based integration (Yan et al., 2021). Neural diffusion processes (NDPs) are another type of diffusion process that extend diffusion models to Gaussian processes, describing distributions over functions with observable inputs and outputs (Dutordoir et al., 2022). Discrete stochastic diffusion processes (DSDPs) view multivariate time series data as values from a continuous underlying function (Biloš et al., 2022). Unlike traditional diffusion models, which operate on vector observations at each time point, DSDPs inject and remove noise using a continuous function. D3VAE is yet another diffusion-based model for sequences (Li et al., 2022a). It starts by employing a coupled diffusion process for data augmentation, which aids in creating additional data points and reducing noise. The model

then utilizes a BVAE alongside denoising score matching to further enhance the quality of the generated samples. Finally, TSGM is a diffusion-based approach to sequence modeling that uses three neural networks to generate sequences (Lim et al., 2023). An encoder is trained to map the underlying time series data into a latent space. Subsequently, a conditional score-matching network samples the hidden states, which a decoder then maps back to the sequence. This methodology enables TSGM to generate good sequences. All these diffusion-based methods lack low-dimensional dynamical latent variables and are slow to sample from as they often rely on Langevin dynamics.

Action Matching (AM) is a method that learns the continuous dynamics of a system from snapshots of its temporal marginals, using cross-sectional samples that are not correlated over time (Neklyudov et al., 2022). AM allows sampling from a system's time evolution without relying on explicit assumptions about the underlying dynamics or requiring complex computations such as backpropagation through differential equations. However, AM does not have low-dimensional dynamical latent variables, which can limit its applicability.

The current widely used class of dynamical models for modeling sequences are state-space models, particularly Mambas (Gu & Dao, 2023). A Mamba uses a latent dynamical model to capture temporal dependencies and an observation process driven by the latent variables to generate data. Mambas are able to model complex and diverse sequences effectively. However, Mambas have limitations. First, the latent variables in Mambas have the same dimensionality as the data, just as for flows, which leads to big models and increases computational complexity. Second, this same high-dimensionality of the latents reduces interpretability, making extracting meaningful insights about the underlying dynamics challenging.

In contrast to the approaches above, Alternators explicitly alternate between generating observations and latent features over time using two neural networks, the OTN and the FTN, jointly optimized to minimize cross-entropy over the observation and feature trajectories. Alternators have low-dimensional latent variables, which enhances their interpretability and makes them more robust to noise in data. Unlike Mambas, which prioritize expressivity using high-dimensional latent variables, Alternators balance computational efficiency, interpretability, and flexibility, excelling in scenarios where understanding a sequence's low-dimensional dynamics is critical.

## 4 Experiments

We now showcase the capabilities of alternators in three different domains. We first studied the Lorenz attractor, which exhibits complex chaotic dynamics. We found alternators are better at capturing these dynamics than baselines such as VRNN, SRNN, NODE, and Mamba. We also used alternators for neural decoding on three datasets to map brain activity to movements. We found that alternators tend to outperform VRNN, SRNN, NODE, and Mamba. Finally, we show that alternators can produce reasonably accurate sea-surface temperature forecasts while only taking a fraction of the time required by diffusion models and Mambas. For comprehensive details regarding implementation specifics and hyperparameter configurations across each experiment, we refer the reader to the Appendix B (code available at: `https://github.com/vertaix/Alternators`).

### 4.1 Model System: The Lorenz Attractor

The Lorenz attractor is a chaotic system with nonlinear dynamics described by a set of differential equations (Lorenz, 1963). We use the attractor to simulate features $z_{1:T}$, with $\boldsymbol{z}_t \in \mathbb{R}^3$ for all $t \in \{1, \ldots, T\}$ and $T = 400$. We simulate from the Lorenz equations by adding noise variables $\epsilon_1, \epsilon_2, \epsilon_3$ to the coordinates,

$$\begin{aligned}
\dot{z_1}(t) &= \sigma \cdot (z_2(t) - z_1(t)) + \epsilon_1, & \epsilon_1 &\sim \mathcal{N}(0,1) \\
\dot{z_2}(t) &= z_1(t) \cdot (\rho - z_3(t)) - z_2(t) + \epsilon_2, & \epsilon_2 &\sim \mathcal{N}(0,1) \\
\dot{z_3}(t) &= z_1(t) \cdot z_2(t) - \beta \cdot z_3(t) + \epsilon_3, & \epsilon_3 &\sim \mathcal{N}(0,1).
\end{aligned}$$

The parameters $\sigma, \rho, \beta$ control the dynamics. Here we set $\sigma = 10, \rho = 28, \beta = 8/3$ to define complex dynamics which we hope to capture well with alternators. Given the features $\boldsymbol{z}_{1:T}$, we simulated $\boldsymbol{x}_{1:T}$,

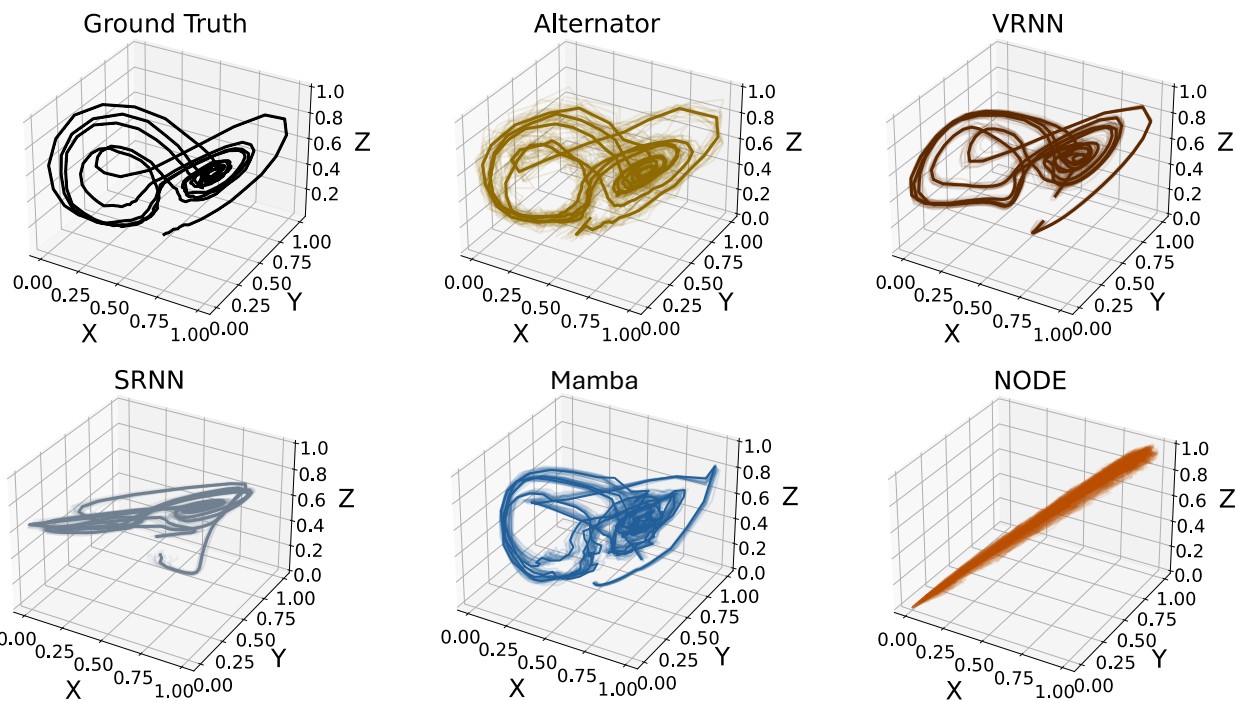

Figure 2: Alternators are better at tracking the chaotic dynamics defined by a Lorenz attractor, especially during transitions between attraction points, than baselines such as VRNN, SRNN, NODE, and Mamba.

Table 1: Alternators outperform several dynamical models in predicting the dynamics defined by the Lorenz equations in terms of MAE, MSE, and CC.

| Method | MAE↓ | MSE ↓ | CC ↑ |
|---|---|---|---|
| SRNN | $0.052 \pm 0.017$ | $0.148 \pm 0.007$ | $0.955 \pm 0.001$ |
| VRNN | $0.074 \pm 0.003$ | $0.173 \pm 0.002$ | $0.963 \pm 0.001$ |
| NODE | $0.044 \pm 0.013$ | $0.220 \pm 0.012$ | $0.888 \pm 0.012$ |
| Mamba | $0.045 \pm 0.003$ | $0.135 \pm 0.001$ | $0.958 \pm 0.001$ |
| Alternator | $\mathbf{0.030 \pm 0.005}$ | $\mathbf{0.076 \pm 0.003}$ | $\mathbf{0.977 \pm 0.001}$ |

with each $\boldsymbol{x}_t \in \{0,1\}^{100}$, by sampling from a time-dependent Poisson point process. We selected the time resolution small enough to ensure $\boldsymbol{x}_t \in \{0,1\}$ for all $t$. We use the Poisson process to mimic spiking activity data. Empirical studies have shown that spike counts within fixed intervals often align well with the Poisson distribution, making it a practical and widely used model in neuroscience (Rezaei et al., 2021; Truccolo et al., 2005). The intensity of the point process is a nonlinear function of the features, $\hat{\lambda}_j(\boldsymbol{z}, t) = \lambda_j(\boldsymbol{z}) * \lambda_{j,H}(t)$, where we define

$$\lambda_j(\boldsymbol{z}) = \exp\left[a_j - \sum_{z_t \in \boldsymbol{z}} \frac{(z_t - \mu_{j,z_t})^2}{2\sigma_{j,z_t}^2}\right] \text{ and } \lambda_{j,H}(t) = \sum_{s_n \in S_j} 1 - \exp\left(-\frac{(t - s_n)^2}{2\sigma_j^2}\right)$$

for $j \in \{1, ..., 100\}$. Here $\mu_{j,z_t}$ and $\sigma_{j,z_t}^2$ are the center and width of the receptive field model of $z_t$, $a_j$ is the maximum firing rate, and $S_j$ is the collection of all the spike times of the $j$th channel. They are drawn from

$$\mu_{j,z_t} \sim U(\mu(z_t) - 2 * \sigma(z_t), \mu(z_t) + 2 * \sigma(z_t)) \tag{16}$$

$$\sigma_{j,z_t} \sim U(\sigma_{min}, 1/100), \ \sigma_j \sim U(\sigma_{min}, 1/100), \text{ and } a_j \sim U(\text{fr}_{min}, \text{fr}_{max}). \tag{17}$$

We set $\text{fr}_{min} = 0$, $\text{fr}_{max} = 10$, and $\sigma_{min} = 0.001$. We then used the paired data $(\boldsymbol{x}_{1:T}, \boldsymbol{z}_{1:T})$ in a sequence-to-sequence prediction task to train an alternator as well as a NODE, an SRNN, a VRNN, and a Mamba.

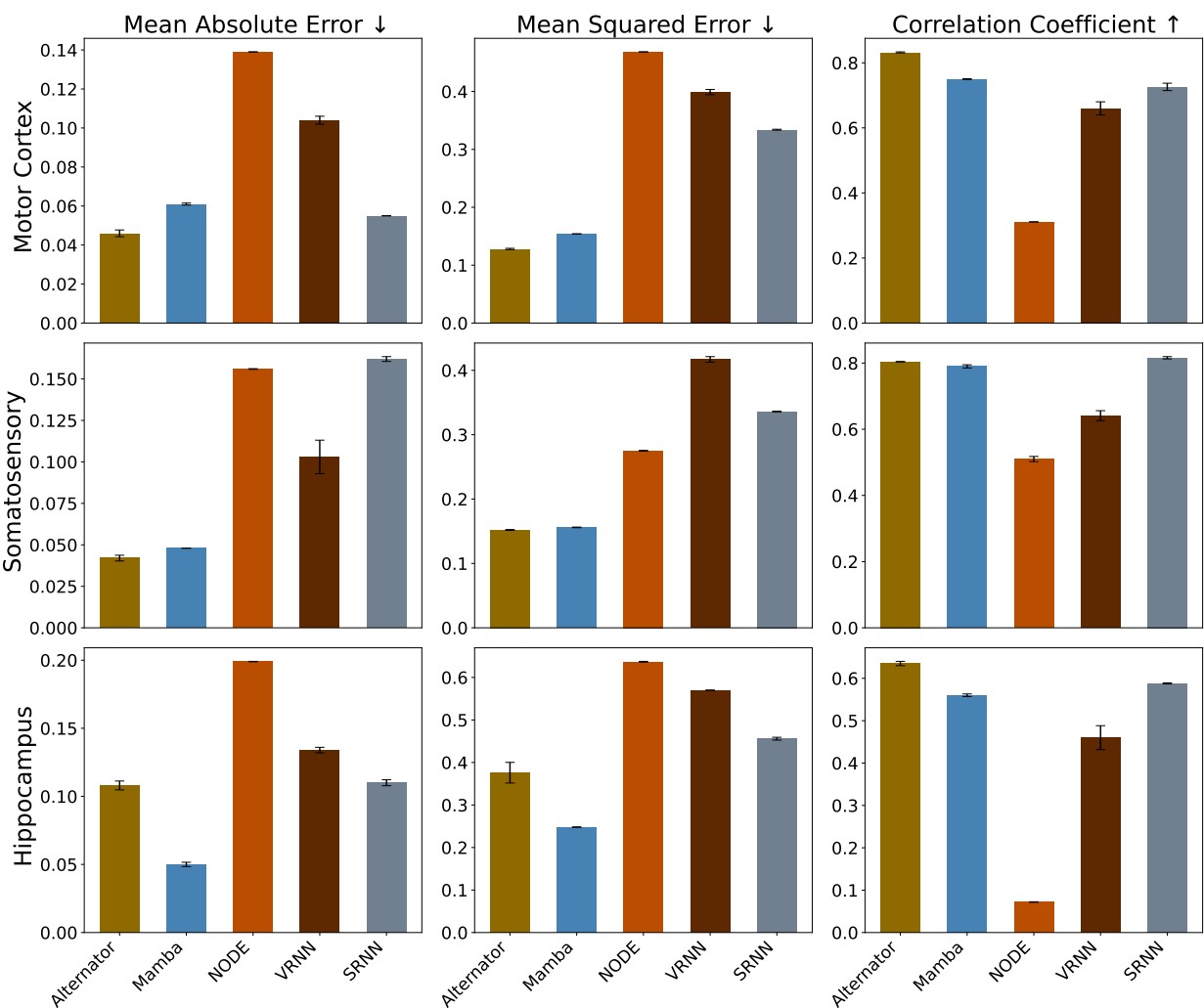

Figure 3: Alternators tend to outperform VRNN, SRNN, NODE, and Mamba on trajectory prediction in the neural decoding task on three different datasets in terms of MAE, MSE, and CC.

We didn't include a diffusion model as a baseline here since it lacks a dynamical latent process that can be inferred from the spiking activities for sequence-to-sequence prediction.

We evaluate each model by simulating 100 new paired sequences following the same simulation procedure. We used the new observations to predict the associated simulated features. We assess feature trajectory prediction performance using three metrics that compare predictions from each model with the ground truth features: Mean Absolute Error (MAE), Mean Squared Error (MSE), and Correlation Coefficient (CC). We used 2-layer attention models, each followed by a hidden layer containing 10 units for both the OTN and the FTN. We set $\sigma_z = 0.1$, $\sigma_x = 0.3$, and $\alpha_t = 0.3$ is fixed for all $t$. The models were trained for 500 epochs using the Adam optimizer with an initial learning rate of 0.01. We applied a cosine annealing learning rate scheduler with a minimum learning rate of 1e-4 and 10 warm-up epochs.

Figure 2 shows the simulated features, along with fits from an alternator, an SRNN, a VRNN, a NODE, and a Mamba. The alternator is better at predicting the true latent trajectory compared to the baselines. Specifically, alternators accurately capture the chaotic dynamics characterized by the Lorenz attractor, especially during transitions between attraction points. The results presented in Table 1 quantify this, with alternators achieving better MAE, MSE, and CC than the baselines.

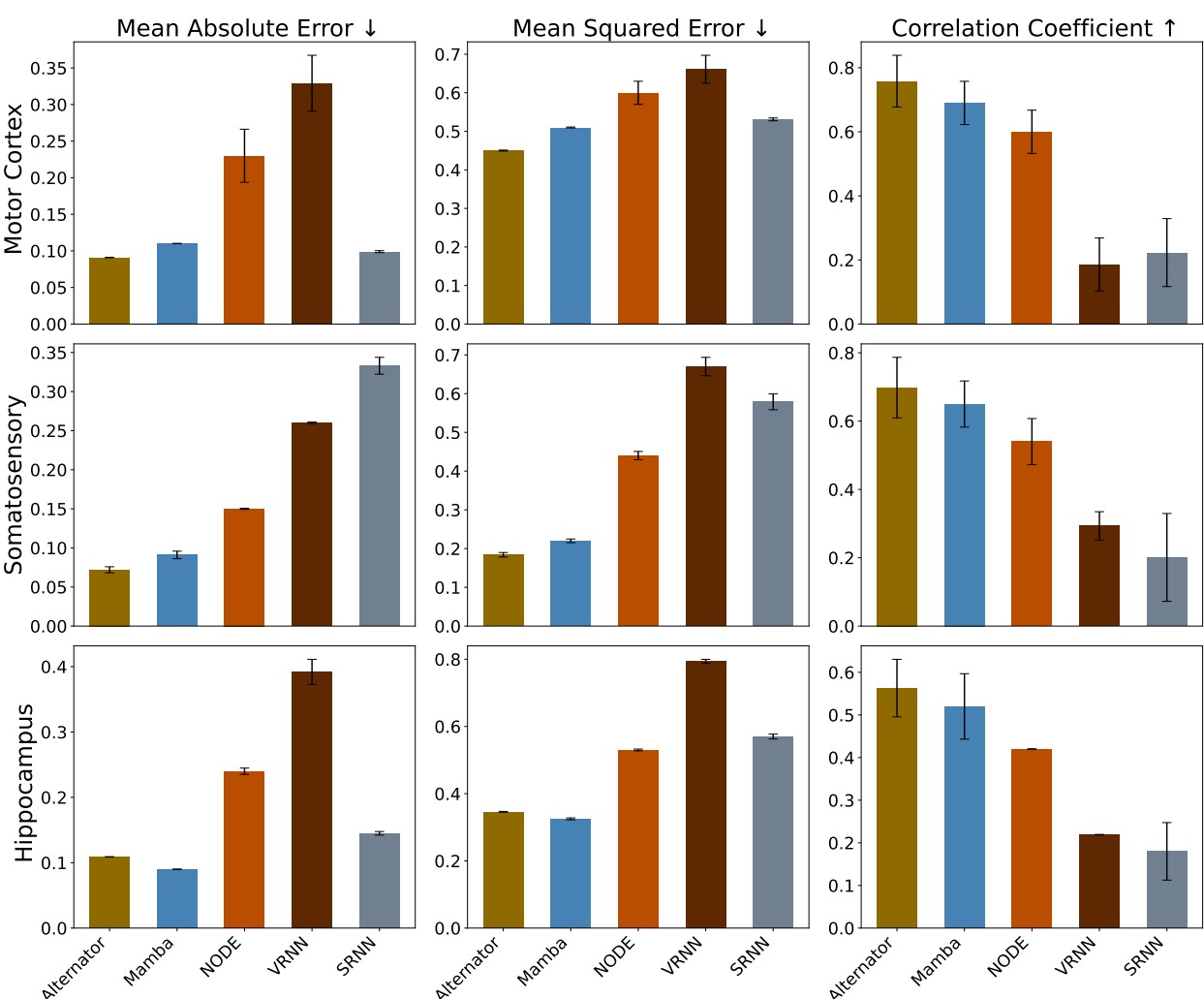

Figure 4: Alternators tend to outperform VRNN, SRNN, NODE, and Mamba on missing value imputation in the neural decoding task on three datasets in terms of MAE, MSE, and CC. The results are averaged across several imputation settings, where we varied the missing value rate from 10% to 95%. The standard errors are shown as vertical bars.

## 4.2 Neural Decoding: Mapping Brain Activity To Movement

Neural decoding is a fundamental challenge in neuroscience that helps increase our understanding of the mechanisms linking brain function and behavior. In neural decoding, neural data are translated into information about variables such as movement, decision-making, perception, or cognitive functions (Donner et al., 2009; Lin et al., 2022; Rezaei et al., 2018; 2023).

We use alternators to decode neural activities from three experiments. In the first experiment, the data recorded are the 2D velocity of a monkey that controlled a cursor on a screen along with a 21-minute recording of the motor cortex, containing 164 neurons. In the second experiment, the data are the 2D velocity of the same monkey paired with recording from the somatosensory cortex, instead of the motor cortex. The recording was 51 minutes long and contained 52 neurons. Finally, the third experiment yielded data on the 2D positions of a rat chasing rewards on a platform paired with recordings from the hippocampus. This recording is 75 minutes long and has 46 neurons. We refer the reader to Glaser et al. (2020; 2018) for more details on how these data were collected. For these experiments, the time horizons were divided into

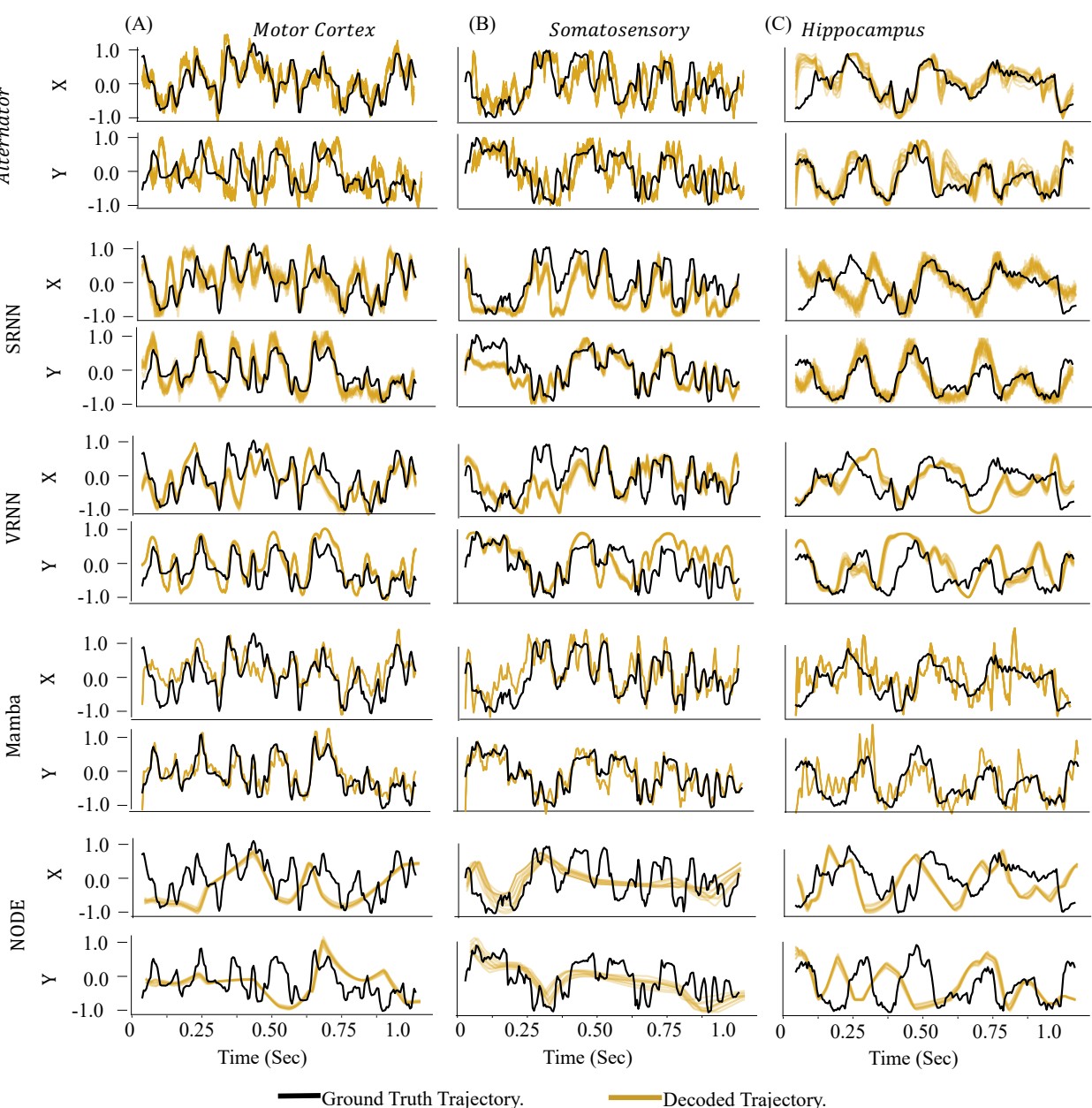

Figure 5: A set of 20 trajectories sampled from different models conditional on spiking activities from datasets: Motor cortex, Somatosensory, and Hippocampus. The alternator produces samples that are closer to the ground truth dynamics.

1-second windows for decoding, with a time resolution of 5 ms. We use the first 70% of each recording for training and the remaining 30% as the test set.

Similarly to the Lorenz experiment, we used attention models comprising two layers, each followed by a hidden layer containing 10 units for both the OTN and the FTN. We set $\sigma_z = 0.1$, $\sigma_x = 0.2$, and $\alpha_t = 0.4$ was fixed for all $t$. The model underwent training for 1500, 1500, and 1000 epochs for Motor Cortex, Somatosensory, and Hippocampus datasets; respectively. We used the Adam optimizer with an initial learning rate of 0.01. We also used a cosine annealing learning rate scheduler with a minimum learning rate of 1e-4 and 5 warm-up epochs.

Table 2: Performance of different models on sea-surface temperature forecasting 1 to 7 days ahead. Numbers are averaged over the evaluation horizon. For SSR, a value closer to 1 is better. The time column represents the time needed to forecast all 7 timesteps for a single batch. Alternators perform reasonably well in terms of CRPS and MSE, and are fast. However, they achieve a worse SSR than MCVD and Dyffusion.

| Method | CRPS $\downarrow$ | MSE $\downarrow$ | SSR $(= 1)$ | Time [s] $\downarrow$ |
|---|---|---|---|---|
| DDPM-P | $0.281 \pm 0.004$ | $0.180 \pm 0.011$ | $0.411 \pm 0.046$ | 0.4241 |
| DDPM-D | $0.267 \pm 0.003$ | $0.164 \pm 0.004$ | $0.406 \pm 0.042$ | 0.4241 |
| DDPM | $0.246 \pm 0.005$ | $0.177 \pm 0.005$ | $0.674 \pm 0.011$ | 0.3054 |
| Alternator | $0.221 \pm 0.031$ | $0.144 \pm 0.045$ | $1.325 \pm 0.314$ | 0.7524 |
| Dyffusion | $0.224 \pm 0.001$ | $0.173 \pm 0.001$ | $1.033 \pm 0.005$ | 4.6722 |
| Mamba | $0.219 \pm 0.002$ | $0.134 \pm 0.003$ | $0.753 \pm 0.009$ | 0.6452 |
| MCVD | 0.216 | 0.161 | 0.926 | 79.167 |

In this experiment, we define the features as the velocity/position and the observations as the neural activity data. We benchmarked alternators against state-of-the-art models, including VRNN, SRNN, NODE, and Mamba on their ability to accurately predict velocity/position given neural activity. We didn't include a diffusion model baseline for the same reason as in the Lorenz experiment, which was also a supervised learning task. We used the same metrics as for the Lorenz experiment. The results are shown in Figure 6, Figure 4, and Figure 5. Alternators are better at decoding neural activity than the baselines on all three datasets.

## 4.3 Sea-Surface Temperature Forecasting

Accurate Sea-Surface Temperature (SST) dynamics prediction is indispensable for weather and climate forecasting and coastal activity planning. Expressivity is important here since prediction performance matters a lot more than interpretability. However, we tested alternators on this task to gauge how they would fare against models such as Mambas and diffusion models on this task. The SST dataset we consider here is the NOAA OISSTv2 dataset, which comprises daily weather images with high-resolution SST data from 1982 to 2021 (Huang et al., 2021). We used data from 1982 to 2019 (15,048 data points) for training, data from the year 2020 (396 data points) for validation, and data from 2021 (396 data points) for testing. We further turned the training data into regional image patches, selecting 11 boxes with a resolution of $60 \times 60$ (latitude $\times$ longitude) in the eastern tropical Pacific Ocean. Specifically, we partitioned the globe into a grid, creating $60 \times 60$ (latitude $\times$ longitude) tiles (Cachay et al., 2023). Eleven grid tiles are strategically subsampled, with a focus on the eastern tropical Pacific region, establishing a refined and consistent dataset for subsequent SST forecasting 1 to 7 days into the future.

We used an ADM (Dhariwal & Nichol, 2021) to jointly model the OTN and the FTN. The ADM is a specific U-Net architecture that incorporates attention layers after each intermediate CNN unit in the U-Net. We selected 128 base channels, 2 ResNet blocks, and channel multipliers of $\{1, 2, 2\}$. We trained the model with a batch size of 10 for 800 epochs, setting $\sigma_z = 0.2$, $\sigma_x = 0.3$, and fixed $\alpha_t = 0.6$ for all $t$. We used the Adam optimizer with an initial learning rate of 0.001 and applied a cosine annealing learning rate scheduler with a minimum learning rate of $1e - 4$ and 5 warm-up epochs.

We compared the alternator against several baselines: DDPM (Ho et al., 2020), MCVD (Voleti et al., 2022), DDPM with dropout enabled at inference time (Gal & Ghahramani, 2016) (DDPM-D), DDPM with random perturbations of the initial conditions/inputs with a fixed variance (DDPM-P) (Pathak et al., 2022), dyffusion (Cachay et al., 2023), and Mamba (Gu & Dao, 2023). We used several performance metrics. One such metric is the Continuous Ranked Probability Score (CRPS) (Matheson & Winkler, 1976), a proper scoring rule widely used in the probabilistic forecasting literature (Gneiting & Katzfuss, 2014; de Bézenac et al., 2020). In addition to CRPS, we also used MSE and Spread-Skill Ratio (SSR). SSR assesses the reliability of the ensemble and is defined as the ratio of the square root of the ensemble variance to the corresponding ensemble RMSE. It serves as a measure of the dispersion characteristics, with values less

than 1 indicating underdispersion (i.e., overconfidence in probabilistic forecasts) and larger values indicating overdispersion (Fortin et al., 2014). We used a 50-member ensemble for each method and compute MSE based on the ensemble mean prediction.

Table 2 shows the results. The alternator achieves reasonable performance in terms of CRPS and MSE, even outperforming Dyffusion and MCVD while being significantly faster. However, the alternator is performing worst in terms of SSR. We attribute this to the alternator's stochasticity, which introduces greater variability into the ensemble predictions.

## 5 Conclusion

We introduced alternators, a new flexible family of non-Markovian dynamical models for sequences. Alternators admit two neural networks that work in conjunction to produce observation and feature trajectories. These neural networks are fit by minimizing the cross-entropy between two joint distributions over the trajectories—the joint distribution defining the model and the joint distribution defined as the product of the marginal distribution of the features and the marginal distribution of the observations, i.e. the data distribution. We showcased the capabilities of alternators in three different applications: the Lorenz attractor, neural decoding, and sea-surface temperature prediction. We found alternators to be stable to train, fast to sample from, and reasonably accurate, often outperforming several strong baselines in the domains we studied.

While alternators demonstrate strong performance across various tasks, several limitations warrant consideration. For imputation tasks, our current approach uses missing-at-random sampling and relies on the latent state $z$ as dynamic memory to generate missing observations in a forward manner. However, bidirectional architectures could enhance performance when both the beginning and end of sequences are available by incorporating backward information flow. Future work could explore integrating bidirectional structures into both the OTN and FTN to smooth predictions through bidirectional processing, potentially improving imputation accuracy for sequences with complex temporal dependencies.

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

## A  Appendix

**Sequence-to-sequence prediction algorithm.** Given paired sequences $\boldsymbol{x}_{1:T}$ and $\boldsymbol{y}_{1:T}$, we employ alternators to predict $\boldsymbol{y}_{1:T}$ from $\boldsymbol{x}_{1:T}$ and vice versa. Algorithm 2 outlines the procedure for sequence-to-sequence prediction using alternators.

**Estimating the log-likelihood of a new sequence.** Sometimes, scientists may be interested in scoring a given sequence using a model fit on data to study how the new input sequence deviates from the data. Alternators provide a way to do this using the log-likelihood. Assume given a new input sequence $\boldsymbol{x}_{1:T}^*$. We

---

**Algorithm 2:** Sequence-To-Sequence Prediction with Alternators

---

Inputs: Samples from $p(\boldsymbol{x}_{1:T}, \boldsymbol{y}_{1:T})$, batch size $B$, $\sigma_x^2$ and $\sigma_y^2$, schedule $\alpha_{1:T}$
Initialize model parameters $\theta$ and $\phi$
**while** *not converged* **do**
    **for** $b = 1, \ldots, B$ **do**
        Draw initial latent $\boldsymbol{z}_0^{(b)} \sim \mathcal{N}(0, I_{D_z})$
        **for** $t = 1, \ldots, T$ **do**
            Compute $\boldsymbol{\mu}_{x_t}^{(b)} = \sqrt{(1 - \sigma_x^2)} \cdot f_\theta(\boldsymbol{y}_{t-1}^{(b)})$
            Compute $\boldsymbol{\mu}_{y_t}^{(b)} = \sqrt{\alpha_t} \cdot g_\phi(\boldsymbol{x}_t^{(b)}) + \sqrt{(1 - \alpha_t - \sigma_y^2)} \cdot \boldsymbol{y}_{t-1}^{(b)}$
        **end**
    **end**
    Compute loss $\mathcal{L}(\theta, \phi)$ in Eq. 14 using samples from $p(\boldsymbol{x}_{1:T}, \boldsymbol{y}_{1:T})$
    Backpropagate to get $\nabla_\theta(\mathcal{L}(\theta, \phi))$ and $\nabla_\phi(\mathcal{L}(\theta, \phi))$
    Update parameters $\theta$ and $\phi$ using stochastic optimization, e.g. Adam.
**end**

---

can estimate its likelihood under the Alternator as follows:

$$\log p_{\theta,\phi}(\boldsymbol{x}_{1:T}^*) = \log \int p_{\theta,\phi}(\boldsymbol{x}_{1:T}^*, \boldsymbol{z}_{0:T}) \; d\boldsymbol{z}_{0:T} \tag{18}$$

$$= \log \int p_{\theta,\phi}(\boldsymbol{z}_{0:T}) \cdot p_\theta(\boldsymbol{x}_1^* \,|\, \boldsymbol{z}_0)) \prod_{t=2}^{T} p_\theta(\boldsymbol{x}_t^* \,|\, \boldsymbol{z}_{t-1})) \tag{19}$$

$$= \log \mathbb{E}_{p_{\theta,\phi}(\boldsymbol{z}_{0:T})} \exp\left[\log p_\theta(\boldsymbol{x}_1^* \,|\, \boldsymbol{z}_0) + \sum_{t=2}^{T} \log p_\theta(\boldsymbol{x}_t^* \,|\, \boldsymbol{z}_{t-1})\right] \tag{20}$$

$$\approx \log \frac{1}{K} \sum_{k=1}^{K} \exp\left[\log p_\theta(\boldsymbol{x}_1^* \,|\, \boldsymbol{z}_0^{(k)}) + \sum_{t=2}^{T} \log p_\theta(\boldsymbol{x}_t^* \,|\, \boldsymbol{z}_{t-1}^{(k)})\right], \tag{21}$$

where $\boldsymbol{z}_{0:T}^{(1)}, \ldots, \boldsymbol{z}_{0:T}^{(K)}$ are $K$ samples from the marginal $p_{\theta,\phi}(\boldsymbol{z}_{0:T})$. Eq. 21 is a sequence scoring function and it can be computed in a numerically stable way using the function logsumexp($\cdot$).

**Neural activity forecasting.** We applied alternators to forecast neural activity across Motor Cortex, Somatosensory Cortex, and Hippocampus datasets, with forecasting rates from 10% to 50%. Evaluated using Mean Absolute Error (MAE), Mean Squared Error (MSE), and Correlation Coefficient (CC), alternators consistently outperform VRNN, SRNN, NODE, and Mamba (Figure 6), demonstrating superior robustness and accuracy across varying forecasting horizons.

# B Implementation Details

This section provides comprehensive implementation details for experimental tasks, including hyperparameter optimization strategies, architectural configurations, and training procedures used across the three main experimental domains.

## B.1 Lorenz Attractor Modeling

For the Lorenz attractor experiments, we employed a 2-layer attention-based architecture for both the Observation Transition Network (OTN) and Feature Transition Network (FTN). Each attention layer was followed by a hidden layer containing 10 units, providing sufficient capacity to capture the complex chaotic dynamics while maintaining computational efficiency.

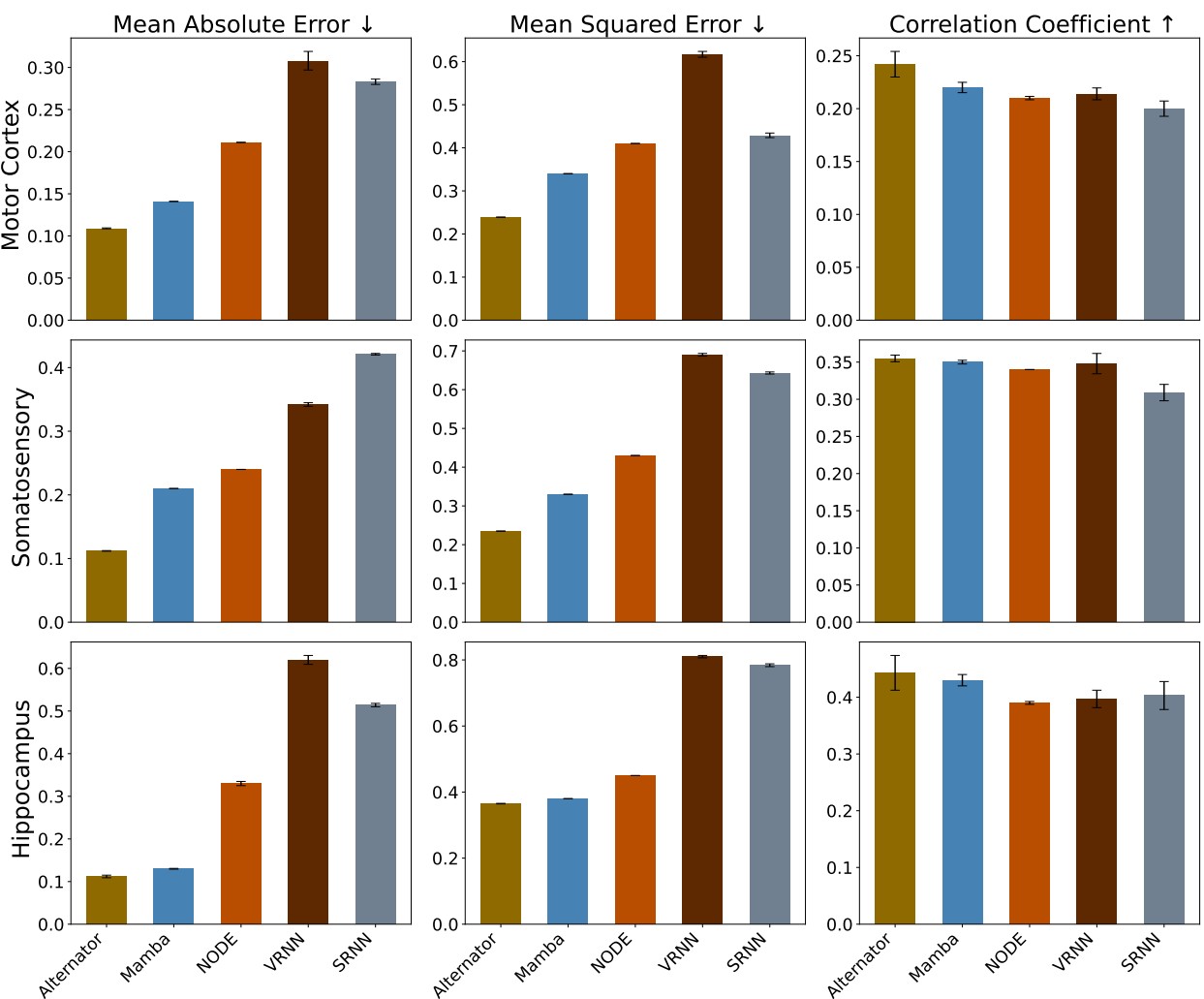

Figure 6: Alternators outperform VRNN, SRNN, NODE, and Mamba on forecasting in the neural decoding task on all three datasets in terms of MAE, MSE, and CC. The results are averaged across several forecasting settings, where we varied the forecasting rate from 10% to 50%. The standard errors are shown as vertical bars.

The noise variance parameters were carefully tuned through grid search optimization $\sigma_z, \sigma_x \in [0.01, 0.8]$. We find the latent noise variance $\sigma_z = 0.1$ and observation noise variance $\sigma_x = 0.3$ as the best choices. The alternation parameter $\alpha_t = 0.3$ was kept fixed across all time steps to maintain consistent switching dynamics between the forward and backward processes. In this experiment, models were trained for 500 epochs using the Adam optimizer with an initial learning rate of 0.01. We applied a cosine annealing learning rate scheduler that reduced the learning rate to a minimum of $1 \times 10^{-4}$ over the training period, with 10 warm-up epochs to stabilize initial training dynamics. The training data consisted of 400 time steps simulated from the Lorenz equations with added Gaussian noise. Model performance was evaluated on 100 newly simulated paired sequences following the same simulation procedure as the training data. We assessed feature trajectory prediction performance using Mean Absolute Error (MAE), Mean Squared Error (MSE), and Correlation Coefficient (CC) metrics, comparing predictions against ground truth Lorenz attractor features.

### B.2 Neural Decoding Experiments

We conducted experiments on three distinct neural datasets. The motor cortex dataset contained 164 neurons recorded over 21 minutes, paired with 2D cursor velocity data. The somatosensory cortex dataset included 52 neurons recorded over 51 minutes, also paired with 2D velocity measurements. The hippocampus dataset comprised 46 neurons recorded over 75 minutes, paired with 2D position data of a rat navigating a reward platform. Time horizons were segmented into 1-second windows for decoding analysis, with a temporal resolution of 5 ms. We allocated the first 70% of each recording for training purposes and reserved the remaining 30% as the test set to ensure robust performance evaluation.

Similar to the Lorenz experiments, we utilized attention models comprising two layers for both OTN and FTN components. Each attention layer was followed by a hidden layer containing 10 units, maintaining architectural consistency across experimental domains while adapting to the specific characteristics of neural data.

Through systematic hyperparameter search, we determined optimal settings for each dataset from a range of $\sigma_z, \sigma_x \in [0.01, 0.8]$. The latent noise variance was set to $\sigma_z = 0.1$, while the observation noise variance was configured as $\sigma_x = 0.2$. The alternation parameter $\alpha_t = 0.4$ was maintained constant across all time steps.

Training epochs were customized for each dataset based on convergence characteristics. The motor cortex dataset required 1500 epochs, the somatosensory dataset also needed 1500 epochs, while the hippocampus dataset converged after 1000 epochs. We employed the Adam optimizer with an initial learning rate of 0.01, coupled with a cosine annealing scheduler that reduced the learning rate to $1 \times 10^{-4}$ with 5 warm-up epochs. Model performance was evaluated using the same metrics as the Lorenz experiments, specifically MAE, MSE, and CC, to assess the accuracy of velocity and position predictions from neural activity patterns.

### B.3 Sea-Surface Temperature Forecasting

We utilized the NOAA OISSTv2 dataset, encompassing daily high-resolution SST data from 1982 to 2021. The dataset was partitioned into training data from 1982 to 2019 (15,048 data points), validation data from 2020 (396 data points), and test data from 2021 (396 data points). We focused on the eastern tropical Pacific Ocean region, extracting 11 boxes with $60 \times 60$ (latitude $\times$ longitude) resolution for detailed analysis.

We employed an ADM network structure to jointly model the OTN and FTN components. The ADM utilized a specialized U-Net architecture incorporating attention layers after each intermediate CNN unit. The configuration included 128 base channels, 2 ResNet blocks per resolution, and channel multipliers of $\{1, 2, 2\}$ to capture multi-scale spatial-temporal patterns effectively.

Models were trained with a batch size of 10 over 800 epochs. The noise variance parameters were set to $\sigma_z = 0.2$ and $\sigma_x = 0.3$, with a fixed alternation parameter $\alpha_t = 0.6$ across all time steps. We used the Adam optimizer with an initial learning rate of 0.001, applying a cosine annealing scheduler that reduced the learning rate to $1 \times 10^{-4}$ with 5 warm-up epochs.

The models were designed to predict SST dynamics 1 to 7 days into the future, providing short to medium-term forecasting capabilities essential for weather and climate applications. Performance assessment utilized multiple complementary metrics. The Continuous Ranked Probability Score (CRPS) served as the primary probabilistic forecasting metric, supplemented by Mean Squared Error (MSE) for deterministic accuracy. The Spread-Skill Ratio (SSR) was employed to evaluate ensemble reliability, with values closer to 1 indicating optimal calibration between forecast uncertainty and actual forecast skill. All SST experiments were conducted on NVIDIA A6000 GPUs with 48GB of memory, enabling efficient processing of the high-dimensional spatial-temporal inputs essential for accurate SST forecasting. All experiments were conducted using appropriate computational resources to ensure reproducible results and fair comparison with baseline methods. Timing measurements were recorded to assess computational efficiency alongside prediction accuracy.

