# OpenReview forum: "Alternators For Sequence Modeling"
_TMLR — Accepted by TMLR_

### Review · Reviewer_eog1 · 2025-04-12

**Summary Of Contributions:**

In this paper, authors proposed a new family of dynamical sequential model called Alternators. By using two separate networks to alternatively generate observations and latent features, authors claimed that the model can better capture the latent dynamics underlying complex data. Through experiments on different datasets, the proposed method has proven to be effective.

**Audience:**

Yes

**Broader Impact Concerns:**

No.

**Claims And Evidence:**

No

**Requested Changes:**

I think authors should spend more efforts on demonstrating their design reasonability and distinguishing with those stochastic differential models.

**Strengths And Weaknesses:**

Strengths:
1. The proposed method was introduced in detail and authors also provided necessary mathematical proofs.
2. Authors conduct extensive experiments on different domains along with sufficient analysis, which made the results more convincible.
3. The paper is well organized, and the writing is relatively clear.

Weaknesses:
1. The motivation of the model design is not fully convincible. According to authors description, $x_{t}$ is derived from $z_{t-1}$ ($x_{t} \sim F(z_{t-1})$), and then $z_{t}$ is obtained from both $x_{t}$ and $z_{t-1}$ ($z_{t} \sim G(z_{t-1}, F(z_{t-1}))$), why the path should be $z_{t-1}$ - $x_{t}$ - $z_{t}$ instead of directly deriving $z_{t}$ from $z_{t-1}$ and decoding the latent $z_{t}$ to $x_{t}$?  In some previous works, the observation $x$ participates into the propagation of latent $z$ to correct the latent feature with given ground-truth, but in authors setting, $x$ doesn’t bring any additional ground-true information.
2. In related works, authors claimed that there is no stochasticity in neural ODE. But there are already some works related to neural SDE, which is a step further beyond ODE by taking stochastic terms into the propagation. Moreover, besides those numerical solvers, there are also simple implementations such as using Euler method, which is in an auto-regressive manner, just like authors model does.

---

> ### Author Response · Authors · 2025-04-12
>
> Dear Reviewer,
>
> Thank you for taking the time to review our manuscript on Alternators. We appreciate your recognition of our detailed mathematical presentation, extensive experimental validation, and clear organization. We would like to address your concerns and highlight some key aspects that may have been unclear in our paper.
>
> **Regarding Model Design Motivation**
>
> The reviewer questions why we use the path $z_{t-1} \rightarrow x_t \rightarrow z_t$ instead of directly deriving $z_t$ from $z_{t-1}$ and then decoding $x_t$ from $z_t$. This concern reflects a misunderstanding of our model's purpose and design.
> In our alternator framework, $x_t$ is not merely a derived value from $z_{t-1}$ without additional information. Rather, this pathway is fundamental to how alternators model real-world dynamics. Many natural processes alternate between hidden dynamics and observable effects. The path $z_{t-1} \rightarrow x_t \rightarrow z_t$ mirrors this structure, where latent states produce observable effects, which then inform the next latent state.
>
> The reviewer states that "x doesn't bring any additional ground-truth information," but this is incorrect. During training, alternators explicitly use ground-truth data for $x_t$, which provides a crucial signal for updating $z_t$. This is critical for learning accurate dynamics. Furthermore, our model is specifically designed to handle both scenarios where ground-truth is available or missing. As demonstrated in our experiments, alternators excel at both imputation (where some ground-truth values are missing) and forecasting (where future ground-truth values are unavailable). This dual capability highlights the flexibility of our design and the importance of incorporating observations into the latent dynamics when available.
>
> The alternating structure enables our model to gain information from observations that would be lost in a direct $z_{t-1} \rightarrow z_t$ path. This is evidenced by our strong performance on chaotic systems like the Lorenz attractor. Our framework minimizes cross-entropy between joint distributions (Eq. 8), which leads to minimizing the entropy of the latent trajectory while ensuring the latent variables yield plausible sequences. This creates a mathematically sound balance between latent space compactness and observational accuracy.
>
> **Regarding Neural SDEs and Stochasticity**
>
> The reviewer correctly points out that neural SDEs exist as extensions to neural ODEs. We acknowledge this omission in our related work section. However, alternators differ fundamentally from neural SDEs in several important ways.
>
> Alternators use direct neural network mappings rather than expensive numerical solvers, making them significantly faster, as demonstrated in our sea-surface temperature forecasting experiments. They explicitly model low-dimensional latent variables ($D_z << D_x$), unlike neural SDEs, which typically maintain high-dimensional state representations. The alternating OTN/FTN architecture allows for specialized network designs tailored to observation and feature space, respectively, which is not possible in neural SDE formulations.
>
> Our cross-entropy minimization approach differs fundamentally from the training objectives used in neural SDEs, leading to better trajectory modeling as shown in our Lorenz attractor experiments. Alternators can be easily applied to sequence-to-sequence prediction, imputation, and forecasting with minimal modifications, while neural SDEs require substantial adaptation for these tasks.
>
> We hope these clarifications assist the reviewer and help highlight the unique contributions of our work. We will revise the manuscript accordingly by incorporating these clarifications and distinctions in the appropriate sections.

---

### Review · Reviewer_XceT · 2025-04-12

**Summary Of Contributions:**

The paper introduces alternators, a novel framework for sequence modeling featuring two neural networks (OTN and FTN) that alternate between generating observations and low-dimensional latent features. Key contributions include:
- A flexible architecture for both generative and sequence-to-sequence tasks, enabling efficient sampling, imputation, and forecasting.
- A training objective based on cross-entropy minimization over joint distributions of observations and latent trajectories, promoting interpretable latent dynamics.
- Empirical validation across three domains: chaotic systems (Lorenz attractor), neural decoding, and climate forecasting, demonstrating superior performance over baselines like Mambas, neural ODEs, and diffusion models in accuracy and speed.

**Audience:**

Yes

**Broader Impact Concerns:**

The paper lacks a broader impact statement.

**Claims And Evidence:**

Yes

**Requested Changes:**

Critical for acceptance:
1. Clarify the theoretical motivation for the loss function (Eq. 7) and its relationship to variational inference or other frameworks.
2. Address the SSR issue in SST forecasting with analysis or adjustments to the model’s stochasticity.

Non-critical but recommended:
1. Provide details on architecture/hyperparameter selection (e.g., why fixed αₜ values, attention layers).
2. Test alternators on non-synthetic latent dynamics or larger-scale datasets.

**Strengths And Weaknesses:**

Strengths:
- The alternator framework introduces a unique alternating mechanism between observation and feature spaces, balancing flexibility and interpretability.
- Validated on diverse tasks (chaotic dynamics, neuroscience, climate) with consistent improvements over strong baselines.
- Faster sampling than diffusion models and competitive runtime compared to Mambas.

Weaknesses:
- The justification for the cross-entropy loss (Eq. 7) lacks rigorous connection to established objectives (e.g., ELBO).
- The Lorenz experiment uses simulated data, raising questions about generalizability to real-world latent dynamics.
- Poor SSR in SST forecasting (Table 2) is attributed to stochasticity but not analyzed or mitigated.

---

> ### Author Response · Authors · 2025-04-16
>
> Thank you for your review of our manuscript on alternators. We greatly appreciate your recognition of our proposed alternating mechanism between the observation and feature spaces, as well as your acknowledgment of the empirical validation across multiple domains. We address your comments below and provide additional clarifications to strengthen the theoretical and practical contributions of our work.
>
> **1. Theoretical Motivation for the Loss Function (Eq. 7)**
>
> We did discuss the distinction between our objective and the traditional ELBO (lines 113-140). Our approach diverges from standard VAE frameworks by explicitly modeling the dynamics underlying the data rather than focusing solely on compressing representations.
> The cross-entropy minimization between the joint distribution and the product of marginals in our loss function has clear information-theoretic motivations:
>
> Entropy Term (Eq. 9): This encourages the latent trajectories to retain maximal information about the dynamics, thereby producing richer representations aligned with the underlying system behavior.
>
> KL Divergence Term: This component encourages the generative model to produce sequences that closely follow the true data distribution, promoting plausibility in the reconstructed trajectories.
>
> This formulation enables alternators to better capture temporal dependencies, particularly in complex dynamical systems, as evidenced by our empirical results. For example, in the Lorenz attractor experiment (see Table 1), the alternator demonstrates superior tracking of chaotic trajectories compared to dynamical VAEs (VRNN, SRNN).
>
> **2. Architectural Choices and Hyperparameter Selection**
>
> We appreciate your questions regarding architectural decisions:
>
> Fixed αₜ Values: While our model allows for time-varying αₜ, we observed that fixed values yielded stable and strong performance across tasks. This choice simplifies training and inference while maintaining the model’s expressive power. Nonetheless, time-varying αₜ remains a viable option for domains where the noise profile is known or can be inferred.
>
> Inclusion of Attention Layers: Attention mechanisms enhance the model’s capacity to capture long-range temporal dependencies, which is particularly advantageous in modeling chaotic or complex biological signals.
>
> Hyperparameter Choices (σₓ and σᵤ): These were selected to strike a balance between reconstruction fidelity and generalization. We conducted sensitivity analyses to ensure robustness across datasets.
>
> **3. Generalizability to Real-World Latent Dynamics**
>
> While the Lorenz attractor experiment involves simulated data, the neural decoding task demonstrates the model’s applicability to real-world scenarios. Across all datasets, alternators consistently outperformed strong baselines, underscoring their robustness and generalizability to diverse, dynamic environments.
>
> **4. Broader Impact**
>
> We agree that a broader impact statement would strengthen the paper. Alternators have potential applications in healthcare (patient monitoring), climate science (improved forecasting), neuroscience (brain-computer interfaces), and anomaly detection systems. These applications could lead to positive societal impacts through improved prediction capabilities for critical systems, though careful consideration of uncertainty quantification would be needed when deployed in high-stakes domains.
>
> Thank you for your constructive feedback. We hope these clarifications address your concerns and strengthen the case for our contributions.

---

### Review · Reviewer_2iWE · 2025-04-15

**Summary Of Contributions:**

The work presents a novel family of models for sequential data. It comprises a pair of learnable neural networks, one (OTN) of which maps from observation space to abstract feature space and the other (FTN) does the opposite. These are trained jointly by minimizing the a cross-entropy type objective over the observation space and the feature space.

Authors use this framework in a couple settings. First, they learn an artificial nonlinear mapping of Lorentz equation trajectories. Second, they learn a real-world mapping between movement and brain activity. Finally, they evaluate alternators on a sea-surface temperature. Overall the performance is satisfactory compared to baselines.

**Audience:**

Yes

**Broader Impact Concerns:**

No concern

**Claims And Evidence:**

Yes

**Requested Changes:**

## Critical

**[RC1]** Attention-type baselines (Transformer, or conformer). See [W3]

**[RC2]** Description of how the hyperparameter were obtained (Could be in the appendix section). See [W5,W6]

## Recommendation

**[RC3]** Derivation for eq (9) in the appendix. See [W1]

**[RC4]** Either a limitation section or limitation discussion in the capabilities sub-sections of Section 2. See [W2]

**[RC5]** Amend related works section to include neural stochastic differential equations. See [W4]

**Minor**
* Some acronyms are sometimes not defined (e.g., SSR and SST in Section 4.3, MAE, MSE, and CC in Section 4.1). I do agree that these are either standard or easily inferable, but it would improve the reading experience.

**Strengths And Weaknesses:**

## Strengths
* The paper is well written and has a clear story
* Related works and baselines covers diverse set of the literature, which I appreciate
* The approach is an interesting twist on the recurrent modeling architecture and has competitive performance.

## Weaknesses

**[W1]** Some derivation would benefit from showing some derivation, notably eq. (9). Note: in eq. (8), is the Bayes rule really used or is it just the conditional probability definition?

**[W2]** The manuscript lacks a bit of discussion on the limitation of the approach. A concrete example is in the 'Imputation and forecasting' use case: because the alternator is a sequential model, it might be limited in settings where data is missing in the middle of a sequence, but the start and the end of the sequence is available. A model with an internal hidden state, e.g., LSTM, or an attention-type model would be able to leverage signal from the end of the sequence for inferring the missing data. Other limitation I might not thinking of now, but the authors might have came across during experimentation would be worth some discussion also.

**[W3]** It seems attention-type architecture (Transformers, Conformer) is missing from related works and baselines. Is there a reason for this? I hate to ask you to add another baseline to your already extensive set of baseline, however they have proven to be very useful in a wide range of task, I would expect them to be competitive in this setting too. See [1,2].

**[W4]** In the related works section authors mentions that a weakness of the Neural ordinary differential equation (NODEs) is that NODEs suffers from modeling noisy observations because they are deterministic. See [3,4].

**[W5]** In the evaluation settings, the authors list the hyperparameters, e.g., $\sigma_z$, $\sigma_x$, $\alpha_t$, lr, ..., used in the experiments. However, it is not clear how the authors arrived at these values. I would be good for a bit of details about what was explored, even is very high level. Hyperparameters obtained by manual exploration of the authors is fine also, lets just mention it here.

**[W6]** Again regarding the hyperparameters: were all the baselines ran with the same set of hyperparameters (lr, warmup period, etc.)? If so, were they the optimal setting for each?

## References
[1] Vaswani, Ashish, et al. "Attention is all you need." Advances in neural information processing systems 30 (2017).

[2] Gulati, Anmol, et al. "Conformer: Convolution-augmented transformer for speech recognition." arXiv preprint arXiv:2005.08100 (2020).

[3] Oganesyan, Viktor, Alexandra Volokhova, and Dmitry Vetrov. "Stochasticity in neural odes: An empirical study." arXiv preprint arXiv:2002.09779 (2020).

[4] Oh, YongKyung, Dong-Young Lim, and Sungil Kim. "Stable neural stochastic differential equations in analyzing irregular time series data." arXiv preprint arXiv:2402.14989 (2024).

---

> ### Author Response · Authors · 2025-04-16
>
> Thank you for your review. We appreciate your positive feedback on the clarity of our presentation and the comprehensiveness of our related work section. Below, we address each of your requested changes:
>
> **[RC1] Attention-type baselines**
>
> Regarding the utilization of attention-based models as baselines, we note that several of our baseline models (including Mamba and the diffusion-based models in the SST section) already incorporate attention mechanisms in their architectures. While we did implement attention components within our Alternator (as mentioned in our experimental setup: "2-layer attention models"), we didn't explicitly compare against standalone Transformer or Conformer architectures.
>
> Prior work, particularly the Mamba [1] paper, has demonstrated substantial improvements over vanilla attention models in sequence modeling tasks.
>
> [1] Gu, Albert, and Tri Dao. "Mamba: Linear-time sequence modeling with selective state spaces." arXiv preprint arXiv:2312.00752 (2023).
>
> **[RC2] Hyperparameter selection methodology**
>
> You're right that we should clarify our hyperparameter selection process. We performed a grid search over key hyperparameters (σ_z, σ_x, α_t) for the Alternator, and manually tuned other hyperparameters like learning rate and architecture depth based on validation performance. For the baselines, we initially used recommended configurations from their original papers, then fine-tuned them on validation data. We'll add a detailed section in the appendix describing this process for all models to ensure fair comparison.
>
> **[RC3] Derivation for equation (9)**
>
> Thank you for pointing this out. You're correct that equation (8) uses the definition of conditional probability rather than Bayes' rule. We'll clarify this and provide a detailed derivation of equation (9) in the appendix of our revised manuscript.
>
> **[RC4] Limitations discussion**
>
> For the imputation task, we used missing-at-random sampling, which allows missing values to occur anywhere in the sequence, including the middle. In this scenario, the latent state z functions as a dynamic memory that helps generate the missing observations.
> We agree that bidirectional architectures can enhance performance when both the beginning and end of a sequence are available. In our revised manuscript, we will discuss how bidirectional structures could be incorporated into both the OTN and FTN to smooth predictions through backward information flow.
>
> We will add a dedicated limitations section discussing constraints of our approach, computational trade-offs, sensitivity to hyperparameter choices, and challenges in extremely high-dimensional settings.
>
> **[RC5] Include neural stochastic differential equations**
>
> We agree that neural stochastic differential equations (NSDEs) deserve mention when discussing limitations of NODEs. We'll expand our related work section to include NSDEs, citing the papers you suggested and discussing how they address the deterministic limitations of NODEs when modeling noisy observations.
>
> Alternators differ fundamentally from neural SDEs in several important ways:
>
> Alternators use direct neural network mappings rather than expensive numerical solvers, making them significantly faster, as demonstrated in our sea-surface temperature forecasting experiments.
>
> They explicitly model low-dimensional latent variables (D_z << D_x), unlike neural SDEs, which typically maintain high-dimensional state representations.
>
> The alternating OTN/FTN architecture allows for specialized network designs tailored to observation and feature spaces, which is not possible in neural SDE formulations.
>
> Our cross-entropy minimization approach differs fundamentally from the training objectives used in neural SDEs, leading to better trajectory modeling as shown in our Lorenz attractor experiments.
>
> Alternators can be easily applied to sequence-to-sequence prediction, imputation, and forecasting with minimal modifications, while neural SDEs require substantial adaptation for these tasks.
>
> **Minor Changes**
>
> We will define all acronyms upon first use, including SSR (Spread-Skill Ratio), SST (Sea-Surface Temperature), MAE (Mean Absolute Error), MSE (Mean Squared Error), and CC (Correlation Coefficient).
> Thank you again for your constructive feedback, which will help us improve our manuscript.

---

### Decision · Action_Editor_Fzxa · 2025-05-19

**Recommendation:** Accept with minor revision

**Comment:**

All reviewers agree that  the claims made in the submission supported by accurate, convincing and clear evidence, and that at least some individuals in TMLR's audience be interested in knowing the findings of this paper.
Two reviewers suggest acceptance, while Reviewer eog1 suggests rejection.

While I agree with Reviewer eog1 that neural SDEs could have been an interesting baseline, there was no explicit request from the reviewer to run further experiments with this baseline, neither in the original review, nor between the authors' reply (12 Apr 2025) and the reviewer's recommendation (15 May 2025).

Authors, please update your manuscript with all promised changes before publication.

**Audience:**

Yes

**Claims And Evidence:**

Yes.

---

> ### Author Response · Authors · 2025-05-24
>
> Thank you for taking the time to review our work. In response, we have incorporated the requested revisions in the camera-ready version, including a discussion on the difference between Alternators and neural SDEs in the related work, added broader impact and limitations, and detailed clarification, implementation, and hyperparameters for Alternators in Appendix B.